# Deep Learning Based HPV Status Prediction for Oropharyngeal Cancer Patients

**DOI:** 10.3390/cancers13040786

**Published:** 2021-02-13

**Authors:** Daniel M. Lang, Jan C. Peeken, Stephanie E. Combs, Jan J. Wilkens, Stefan Bartzsch

**Affiliations:** 1Institute of Radiation Medicine, Helmholtz Zentrum München, 85764 Munich, Germany; jan.peeken@tum.de (J.C.P.); stephanie.combs@tum.de (S.E.C.); stefan.bartzsch@tum.de (S.B.); 2Department of Radiation Oncology, School of Medicine and Klinikum Rechts der Isar, Technical University of Munich (TUM), 81675 Munich, Germany; wilkens@tum.de; 3Physics Department, Technical University of Munich, 85748 Garching, Germany; 4Deutsches Konsortium für Translationale Krebsforschung (DKTK), Partner Site, 81377 Munich, Germany

**Keywords:** HPV status, oropharyngeal cancer, deep learning, transfer learning, machine learning

## Abstract

**Simple Summary:**

Determination of human papillomavirus (HPV) status for oropharyngeal cancer patients depicts a essential diagnostic factor and is important for treatment decisions. Current histological methods are invasive, time consuming and costly. We tested the ability of deep learning models for HPV status testing based on routinely acquired diagnostic CT images. A network trained for sports video clip classification was modified and then fine tuned for HPV status prediction. In this way, very basic information about image structures is induced into the model before training is started, while still allowing for exploitation of full 3D information in the CT images. Usage of this approach helps the network to cope with a small number of training examples and makes it more robust. For comparison, two other models were trained, one not relying on a pre-training task and another one pre-trained on 2D Data. The pre-trained video model preformed best.

**Abstract:**

Infection with the human papillomavirus (HPV) has been identified as a major risk factor for oropharyngeal cancer (OPC). HPV-related OPCs have been shown to be more radiosensitive and to have a reduced risk for cancer related death. Hence, the histological determination of HPV status of cancer patients depicts an essential diagnostic factor. We investigated the ability of deep learning models for imaging based HPV status detection. To overcome the problem of small medical datasets, we used a transfer learning approach. A 3D convolutional network pre-trained on sports video clips was fine-tuned, such that full 3D information in the CT images could be exploited. The video pre-trained model was able to differentiate HPV-positive from HPV-negative cases, with an area under the receiver operating characteristic curve (AUC) of 0.81 for an external test set. In comparison to a 3D convolutional neural network (CNN) trained from scratch and a 2D architecture pre-trained on ImageNet, the video pre-trained model performed best. Deep learning models are capable of CT image-based HPV status determination. Video based pre-training has the ability to improve training for 3D medical data, but further studies are needed for verification.

## 1. Introduction

Infection with the human papillomavirus (HPV) has been identified as oncogenic for several cancer sites [1] and chronic HPV infections may also lead to oropharyngeal cancer (OPC) development [2]. While smoking and alcohol consumption, two well established risk factors for OPC, have notably declined in North America and Northern Europe [3], infections with HPV have increased and lead to growing incidence rates of OPC [1]. Plummer et al. [4] estimated in 2016 that around 31% of OPC cases globally are caused by HPV. However, OPC patients with a positive HPV status show a 74% reduced risk of cancer related death [5] and HPV-positive tumors are more radiosensitive than HPV-negative tumors. Hence, determination of the HPV status has become an essential diagnostic factor, and dose de-escalation studies, such as the ECOG 1308 trial [6], seek to reduce therapy induced side effects for HPV-positive OPC patients. The detection of HPV-induced overexpression of p16^INK4a^ by immunohistochemistry is most frequently used with a reported sensitivity of >90% and a specificity of >80% [7,8].

Machine learning techniques, in combination with high-dimensional personalized “-omics” datasets, have shown to be powerful tools for the prognostic and predictive assessment of therapeutic efficacy [9,10,11]. Radiomics refers to the extraction of information from radiological images by applying hand-crafted filters on preselected regions of interest [12]. Segal et al. [13] demonstrated that features in radiological images can be used to reconstruct the majority of the tumor genetic profile. Radiomics data successfully predicted overall survival [14,15], metastases development [16,17] or histological properties [18,19] and may be used as a decision support system in clinical practice. Radiomics approaches to determine the HPV status achieved areas under the receiver operating characteristic curve (AUC) of about 70% to 80%, when tested on external data sets [20,21].

Although radiomics has proven its potential in medical image analysis, deep learning was shown to be clearly superior in most other computer vision tasks. Deep learning features an end-to-end training without the need to design hand-crafted filters. However, training requires considerably larger patient numbers than typically available in clinical applications. One possible solution to this problem is given by transfer learning, an approach making use of large non-medical data sets in order to inject information into the network before the actual learning task is started. Fujima et al. [22] trained a 2D convolutional neuronal network (CNN) on FDG-PET images to classify HPV status in OPC patients and achieved an AUC of 83%, using a transfer learning approach based on natural images from the ImageNet database [23]. However, they did not test their data on an external cohort and excluded images containing severe motion artifacts and tumors with diameters below 1.5 cm.

In this work, we investigate deep learning on diagnostic CT images as a tool to attribute oropharyngeal cancer to a human papillomavirus-driven oncogenesis.

Our study was based on four different publicly available data sets of The Cancer Imaging Archive (TCIA) [24]. Transfer learning facilitated deep learning on the relatively small data set size on a CNN derived from the C3D classification network [25], with weights pre-training on the Sports-1M data set [23]. Hussein et al. [26] used C3D to initialize a multi-task learning approach for lung nodule risk stratification. However, their network depended on additional information, such as tumor sphericity and texture, while we trained our network in a simple end-to-end fashion.

## 2. Material and Methods

### 2.1. Data

Head and neck cancer collections OPC-Radiomics [27,28], HNSCC [29,30], Head-Neck-PET-CT [31,32] and Head-Neck-Radiomics-HN1 [33,34] of the publicly accessible TCIA archive [24] were mined for appropriate cases. Inclusion criteria were: oropharyngeal subtype, existence of a pre-treatment CT image with respective segmentation of the gross tumor volume (GTV) and detected HPV status. Only the centerpoint of the available GTV was used to cut the images to smaller size, i.e., no exact delineation of the tumor was needed. For the Head-Neck-PET-CT data set, the GTV also involved delineation of lymph nodes. In total, this led to 850 individual oropharyngeal cancer patients (Table 1); example images can be seen in Figure A2 of the Appendix A.

To ensure generalizability to images from institutions and scanners not seen during training, testing on external data is inevitable in a medical setting [35]. Hence, for training, validation and testing independent data sets were employed. The OPC-Radiomics and HNSCC data sets were combined and used as a training set, since these two cohorts contained the most cases. Due to its variety in cases with data coming from 4 different institutions, Head-Neck-PET-CT was employed for validation. The validation set is used for selection of the final model weights; a versatile validation set therefore supports selection of a model applicable to a broad kind of settings. The Head-Neck-Radiomics-HN1 data set was used as a test set.

The OPC-Radiomics and Head-Neck-Radiomics-HN1 data sets provided the HPV status tested by immunohistochemical (IHC)-based p16 staining. A combination of p16 IHC and/or HPV DNA in situ hybridization was used in the HNSCC data. For Head-Neck-PET-CT, testing methods were not reported.

We resampled all CT images to an isotropic voxel size of 1 mm^3^. Voxel values, given in Hounsfield units, were cropped at −250 HU and 250 HU, and linearly rescaled to integer values ranging from 0 to 255.

### 2.2. Deep Neural Network

Transfer learning is commonly used to overcome the problem of small data set sizes. A widely applied approach uses the ImageNet data set [23], consisting of natural images, for pre-training. However, CT images are 3 dimensional and therefore pre-training should be performed on 3 dimensional data. We tested the capability of video data based pre-training defining the time axis as the 3rd dimension.

To avoid long training times and obtain a reproducible starting point, we used the already trained video classification network C3D [25,36] as a pre-trained base line model. C3D processes video input in a simple 3D convolutional manner, i.e., all three input dimensions are handled in the same way. Convolutional layers are followed by max-pooling layers, ending with three densely connected layers and a softmax activation layer. C3D was trained to predict labels for video clips of the sport-1M data set [37], which contains 1.1 million videos of sport activities belonging to one of 487 categories. For training, 16 image frames per video clip with a size of 112×112 were used, i.e., input dimension was given by 16×112×112×3.

All densely connected layers were removed from the network and weights of all convolutional layers were frozen during training. New, randomly initialized, densely connected layers were then added after the last convolutional layer, resulting in a single output neuron followed by a sigmoid activation layer. All dense layers were followed by a ReLU activation layer and a dropout layer. The best model consisted of two dense layers with size 1024 and 64, with a dropout rate of 0.35 and 0.25, respectively (Figure 1a).

Weighted binary cross entropy was chosen as loss function with the weights set, such that both classes contributed equally. The Adam optimizer was used with a learning rate of $10^{-4}$ and the batch size was 16.

As input a single image of size 112×112×48 was cut from each of the CT scans and then rearranged to fulfill the input requirements, i.e., 3 consecutive layers were fed to the color channels resulting in information about 16 of those combined layers in longitudinal direction. The data augmentation techniques applied included: flipping on the coronal and the sagittal plane, rotation by a multiple of $90^{\circ }$ and shifting of the GTV center point by a value between 0 and 7 pixels in both directions of the transverse plane.

For comparison, we also trained a 3D convolutional network from scratch, i.e., with all weights randomly initialized, and a 2D network pre-trained on ImageNet.

The general architecture of the model trained from scratch followed that of C3D net. Only the size of the network was reduced to avoid overfitting. To do so, max-pooling was applied after every convolutional layer, except for the second but last one, and dropout was also already applied in the last convolutional layer. The final model consisted of convolutional layers with feature map sizes of 16,32,32,64,128,128, followed by densely connected layers of size 256 and 128 (Figure 1b). Dropout rate for the last convolutional layer and the two dense layers was 0.25. All convolutional kernels were chosen to be of size 3×3×3. In order to not merge the signal in the time dimension too early, the C3D model used a kernel and stride of size 2×2×1 in the very first max-pooling layer. We followed this approach to account for the smaller input size in longitudinal dimension. All other max-pooling kernels were of size 2×2×2 with a stride of the same shape. Input images were cut from the CT scans in exactly the same way as before except for the rearrangement of layers, i.e., input size was 112×112×48. Data augmentation was applied as before and the optimizer and learning rate stayed the same.

For the 2D model, the VGG16 architecture of [38], pre-trained on ImageNet, was chosen. One image was cut from each CT scan in the exact same way as for the 3D model trained from scratch. For training, the images were split into 16 slices of size 112×112×3 to fit the networks input dimensions, i.e., three consecutive slices were fed to the color channels of the network. During testing, the overall prediction score was constructed by averaging the prediction scores of all 16 slices. All dense layers were removed from the model and replaced by randomly initialized layers. Dense layers were again followed by ReLU activations and dropout layers to finally end in one single output neuron with a sigmoid activation function. Weights of all convolutional layers were again kept fixed. The best performing model had a size of 512 followed by 64, with a dropout rate of 0.5 and 0.25, respectively (Figure 1c). Data augmentation was performed as before.

The code has been made publicly available (https://github.com/LangDaniel/hpv_status accessed on 10 February 2021).

## 3. Results

All models were trained for 200 epochs. Training performance is given in Figure A1 of the Appendix A. Weights of the epoch with the best performing loss were chosen for the respective final model. Due to the relative small size of our test and validation sets, we chose to train each model with the same hyper-parameter settings 10 times and report mean results.

The 3D video pre-trained model achieved the highest validation AUC with a mean (min, max) value of 0.73 (0.69, 0.77) and a corresponding training AUC of 0.95 (0.90, 0.98). The 3D network trained from scratch reached a slightly less validation AUC of 0.71 (0.67, 0.74), training AUC was given by 0.83 (0.66, 0.92). Results for the 2D network pre-trained on ImageNet were given by 0.62 (0.58, 0.64) for validation and 0.78 (0.76, 0.79) for training.

Receiver operating characteristics (ROC) results on the test set can be seen in Figure 2.

The test AUC for the video based network was given by 0.81 (0.77, 0.84), for the 3D network trained from scratch and the pre-trained 2D network; test AUCs were given by 0.64 (0.56, 0.70) and 0.73 (0.70, 0.75), respectively.

Sensitivity, specificity, positive and negative predicted values (PPV and NPV respectively) and the $F_{1}$ score, given by the harmonic mean of precision and recall, were computed for a threshold value of 0.50 in the output layer. Test set results are shown in Table 2.

## 4. Discussion

From the three networks trained, the video pre-trained C3D model performed best with a test AUC score showing clear superiority over the two other models.

We associate the success of pre-training with two factors. First of all, transfer learning is counteracting the problem of small data set sizes by injection of knowledge prior to the actual learning task. General benefit of natural imaging based pre-training for the medical domain has been challenged by Raghu et al. [39], accounting improvements only to the application of over-parametrized models. However, our training data set involved just a few hundred cases, making even the training of small networks difficult. We therefore accredit the model improvement, in our case, to actual transferred knowledge between the two domains. Second, transfer learning improves generalization. Different studies have shown that radiomic feature values are affected by CT scanners and scanning protocols [40,41,42,43]. Convolutional neural networks are sensitive to such domain shifts [44,45]. Hendrycks et al. [46] showed that transfer learning can improve model robustness, which leads to better validation and testing results.

Additionally, we recognize improved performance of the 3D approach over the 2D approach. Our results suggest that the third dimension contains essential information for HPV classification. Since the 2D case uses only small parts of the complete CT image, a 3-dimensional input volume enables a better fit to the data. The small input volume increases the probability to receive slices that lack information on patient’s HPV status and impair the training progress. Moreover, single slices can suffer from artifacts, of which Figure A2 gives an impression. Providing only three CT slices increases the impact of such artifacts and compromises training success.

When tested on an external data set, radiomic models reached AUC scores ranging between 0.70 and 0.82 [20,21]. A crucial difference between radiomics and deep learning lies within the input information. Radiomic methods apply filters on a predefined region of interest, typically the GTV, for feature generation. HPV-positive OPCs, however, are known to be associated with regions outside the tumor volume, e.g. cystic lymph nodes [47,48]. Therefore, radiomic approaches usually ignore important parts of the image. Deep learning, on the other hand, does not require the delineation of the tumor volume, leading to an advantage in HPV prediction.

Our work has proven the capability of deep learning models to predict patients’ HPV status based on CT images from different institutions, with a sensitivity and specificity of 75% and 72%, respectively. For clinical application, further studies are required. This involves the implementation of more diverse training data to investigate the impact on model generalizability. Furthermore, the effect of data quality has to be examined. Head and neck CT images are generally prone to artifacts; Leijenaar et al. [49] analyzed a subset of the OPC-Radiomics collection and found that 49% of all cases contained visible artifacts. They showed that the involvement of artifacts plays a major role for HPV status prediction, by training a radiomics model which achieved AUC scores ranging between 0.70 and 0.80 depending on the inclusion of cases with artifacts in the training and test cohorts [50]. Right now, the model is trained predominantly and tested solely on HPV cases determined by p16 IHC. For further validation, other testing methods should be included.

The 8th edition of the American Joint Committee of Cancer (AJCC) staging manual defined HPV-positive OPCs as an independent entity and recommended testing by immunohistochemical staining of p16, a surrogate marker of HPV [51]. However, p16 overexpression is not exclusively linked to an ongoing carcinogenesis caused by a HPV infection, leading to a low specificity of p16 testing with up to 20% of OPC p16-positive cases being HPV-negative [52]. Tests with higher specificity exist, but are technically challenging. The current gold standard test of E6/E7 mRNA detection by PCR remains labor-intensive and, hence, relatively expensive. [53,54]. In a North-American survey, Maniakas et al. [55] found that only about 2/3 of head and neck cancer cases were tested for HPV infection, with cost and time limits as the major reasons not to test. An imaging-based test is easily applicable, non-invasive and also features time efficiency and low cost. Our model could be used for cases where those constraints forbid histopathological testing. In order to serve as a replacement for current methods, sensitivity and specificity of the model have to be further improved. However, in its current state, the model could be used in combination with other testing methods to achieve higher total performance.

## 5. Conclusions

It was demonstrated that convolutional neural networks are able to classify the HPV status of oropharyngeal cancer patients, based on CT images from different cohorts. The sports video clip-based transfer learning approach performed best in comparison to two other CNN models. Video based pre-training has the potential to improve deep learning on 3D medical data, but further studies are needed for general verification.

## Figures and Tables

**Figure 1 cancers-13-00786-f001:**
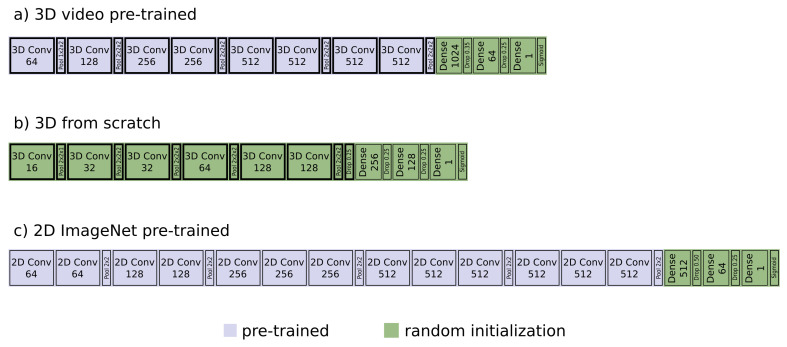
The convolutional neural networks. With (**a**), the network pre-training on the video data of the 1M sports data set. All convolutional layers (Conv) were kept fix, while the fully connected layers (Dense) were replace by randomly initialized ones. (**b**) shows the CNN trained with all layers randomly initialized. Network size was reduced in comparison to (**a**), max-pooling (Pool) was applied earlier and dropout (Drop) was already used in the last convolutional layer. (**c**) the 2D architecture, pre-trained on ImageNet. As base line model VGG16 was used, all convolutional layers were kept fixed, dense layers were replaced and randomly initialized.

**Figure 2 cancers-13-00786-f002:**
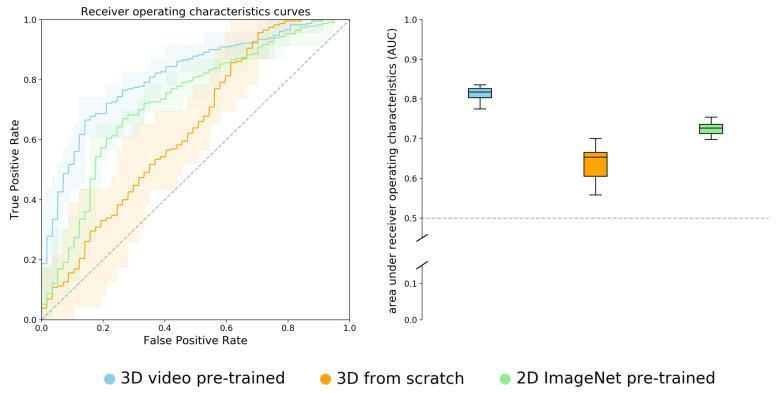
Combined receiver operating characteristics (ROC) plots and AUC score box plots of the test set for the three different models. The ROC plots represent the mean curve of the ten times that models were trained, the shadowed area represents the minimal and maximal curves. Boxes represent lower and upper quartiles, whiskers minimal and maximal values. Median (min, max) AUC score for the video pre-trained model was given by 0.82 (0.77, 0.84). The 3D network trained from scratch and the 2D ImageNet pre-trained model reached an AUC score of 0.65 (0.56, 0.70) and 0.73 (0.70, 0.75), respectively.

**Table 1 cancers-13-00786-t001:** Patient information for the different cohorts. Continuous variables are represented by mean values and ranges by ($q_{25}$–$q_{75}$), with $q_{25}$ and $q_{75}$ being the 25th and 75th percentiles, respectively.

2lClinical Variable		Training Set		Validation Set	Test Set
Cohort		OPC	HNSCC	HN PET-CT	HN1
Number of patients		412	263	90	80
HPV: pos/neg		290/122	223/40	71/19	23/57
HPV status					
Age					
pos		58.81 (52.00–64.75)	57.87 (52.00–64.00)	62.32 (58.00–66.00)	57.52 (52.00–62.50)
neg		64.82 (58.00–72.75)	60.02 (54.50–67.25)	59.11 (49.50–69.50)	60.91 (56.00–66.00)
Sex: Female/Male					
pos		47/243	32/191	14/56	5/18
neg		34/88	15/25	4/15	12/45
T-stage: T1/T2/T3/T4					
pos		46/93/94/57	60/93/41/29	10/37/15/9	4/8/9/8
neg		9/35/43/35	6/12/12/10	3/4/8/4	9/16/9/23
N-stage: N0/N1/N2/N3					
pos		33/22/215/20	19/30/170/4	11/10/47/3	6/2/15/0
neg		36/16/62/8	5/2/31/2	2/1/13/3	14/10/31/2
Tumor size [cm${}^{3}$]					
pos		29.35 (10.52–37.78)	11.78 (3.94–14.04)	34.63 (14.91–41.77)	23.00 (10.83–34.29)
neg		36.99 (15.72–45.35)	23.57 (5.80–22.85)	35.09 (17.32–47.82)	40.19 (11.77–54.42)
transversal voxel spacing [mm]		0.97 (0.98–0.98)	0.59 (0.49–0.51)	1.06 (0.98–1.17)	0.98 (0.98–0.98)
longitudinal voxel spacing [mm]		2.00 (2.00–2.00)	1.53 (1.00-2.50)	2.89 (3.00–3.27)	2.99 (3.00–3.00)
manufacturer					
	GE Med. Sys.	272	238	45	0
	Toshiba	138	3	0	0
	Philips	2	12	45	0
	CMS Inc.	0	0	0	43
	Siemens	0	4	0	37
	other	0	6	0	0

**Table 2 cancers-13-00786-t002:** Test results for the three different models. Scores represent mean (std) values for the ten times each model was trained.

Metric	3D Video Pre-Trained	3D from Scratch	2D ImageNet Pre-Trained
AUC	0.81 (0.02)	0.64 (0.05)	0.73 (0.02)
sensitivity	0.75 (0.06)	0.67 (0.12)	0.84 (0.07)
specificity	0.72 (0.09)	0.49 (0.09)	0.40 (0.13)
PPV	0.53 (0.07)	0.35 (0.03)	0.37 (0.04)
NPV	0.88 (0.02)	0.79 (0.05)	0.87 (0.03)
$F_{1}$ score	0.62 (0.02)	0.45 (0.05)	0.51 (0.03)

## Data Availability

All data used in this study was taken from The Cancer Imaging Archive (TCIA) [24] and is publicly accessible.

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
