# Peer review of "Deep Learning Based HPV Status Prediction for Oropharyngeal Cancer Patients"

_cancers, 2021, doi:10.3390/cancers13040786_

Round 1

Reviewer 1 Report

Thank you for improving the paper. Now it is well organized, methods are adequately described and the discussion is appropriate.

Author Response

Thanks very much for your positive comments.

Reviewer 2 Report

Dear Authors,

Thank you very much for your revision. I think now your claim is clear and looks great. It is quite interesting story, and I hope it would be applied into the real clinical setting in near future.

Author Response

(The authors gave the same response as above.)

Reviewer 3 Report

The authors invested a satisfactory amount of work to respond to the reviewers comments and to justify their methodology and results obtained.

The quality of the manuscript is now significantly improved and I am substantially satisfied.

Some minor issues still remain:

Introduction:

The text has been rearranged according to the reviewer’s suggestion. The number of the references must be consequently modified and numbered in ascending order according to the journal recommendation.

Figure 2 caption:

  • The Median AUC score values for the video pre-trained model and for the 3D network trained from scratch (0.82 and 0.65 respectively) are different from those displayed in the text (lines 177-178) and in table 2.
  • “3D ImageNet pre-trained model” must be modified with “2D ImageNet pre-trained model”

Author Response

Thank you again for the detailed revision of your work that led to real improvement of the manuscript.

1. The text has been rearranged according to the reviewer’s suggestion. The
number of the references must be consequently modified and numbered in
ascending order according to the journal recommendation.
Answer: Done. The manuscript was written in Latex, we simply forgot
to recompile.

2. The Median AUC score values for the video pre-trained model and for the
3D network trained from scratch (0.82 and 0.65 respectively) are different
from those displayed in the text (lines 177-178) and in table 2.Answer: For the caption in Figure 2 the values are given by median (min, max) while in the text they are given by mean (min, max) values. The reason for this is that we followed the common approach to display median values for the box plots but thought that mean values would be more appropriate for the analysis in the text.

3. “3D ImageNet pre-trained model” must be modified with “2D ImageNet
pre-trained model”
Answer: Done. Thank you very much for spotting this.

Sincerely Yours,
Daniel Lang on behalf of all authors.

This manuscript is a resubmission of an earlier submission. The following is a list of the peer review reports and author responses from that submission.

Round 1

Reviewer 1 Report

This is an interesting study about deep learning based HPV status prediction for oropharyngeal cancer patients. The authors tested the ability of deep learning models for HPV status testing based on routinely acquired diagnostic CT images. They found that deep learning models are capable of CT image based HPV status determination.

The paper is well written. However, some issues remain.

The authors stated that different methods for HPV testing were used in different data set. This is a bias for the study and it must be discussed.

Sensibility and specificity of deep learning based HPV status prediction were 75% and 72%, respectively. Since they were inferior to those of p16 staining, the authors should not conclude that “Our work has proven the capability of deep learning models to predict patients HPV status based on CT images from different institutions.” The conclusions should be changed according to the true clinical value of deep learning based method.

Reviewer 2 Report

The authors investigated the deep learning for the prediction of HPV infection in OCC patients. It is quite interesting story, but it is hard to understand the clinical significance of this study. 

  1. Is there any representative images how to determine the HPV status? 
  2. Which types of HPVs are predicted? 
  3. How the prediction would affect the clinical managements?

Reviewer 3 Report

The paper reports the results of a transfer learning strategy to classify differentiate HPV-positive from HPV-negative cases relying on four different publicly available dataset, two of them used for training, one for validation and one for testing. The main novelty stands in the use of a CNN deep network pre-trained on movies, so to use an architecture designed to deal with 3D data. This architecture is considered to be better suited to 3D CT scans. Results are compared with those obtained with a 2D pretrained network (VGG on image data) and with a 3D network designed from scratch.

All in all the methodology to build the network is sound, since training, validation and test data come from different databases, and therefore the potential generalization capability of the learned network is tested, too.

However, there are at least two major issues related to the approach here presented.

  1. The comparison with other strategies is not solid. The two alternative networks (2D and from scratch) achieved performances considerably lower than the 3D pretrained CNN. In general, the AUC reached in the training by the new strategy is way too high than the other two, so that there is a clear difference in the quality of the design of the different approaches. This may be related to a convergence not achieved or a modeling strategy that was non optimized. In particular, for the 2D 0.62 AUC in validation is below acceptability. As a comment, averaging the frames before classification in 2D is probably a choice that leads to a strong loss of information.
  2. The analysis presented does not clarify if the improvement of the results is related to a general capability of 3D pretrained networks to handle 3D CT scans or if this is related to the particular problem at hand. Before taking into account a specific case, which has some peculiarities related to the disease and to the datasets, including the order of the analysis, it would be important to investigate the overall properties of the strategy by using the same approach on other 3D CT datasets.

Finally, a deeper explanation of the classification properties of the method, explaining why there were misclassifications, would be required before publishing in a clinical journal.

Minor comment:
- The introduction chapter sounds rather disorganized and, in my opinion, the text should be restructured as follow (numbers of the references must be consequently changed):

"HPV has been identified oncogenic for several cancer sites [12]. Chronic HPV infections may also lead to OPC development [13]. While smoking and alcohol consumption, two well established risk factor sfor OPC, have notably declined in North America and Northern Europe [14], infections with HPV have increased and lead to growing incidence rates of OPC [12]. Plummer et al. [15] estimated in 2016 that around 31% of OPC cases globally are caused by HPV. However, OPC patients with a positive HPV status show a 74% reduced risk of cancer related death [16] and HPV-positive tumors are more radiosensitive than HPV-negative tumors. Hence, determination of the HPV status has become an essential diagnostic factor and dose de-escalation studies, such as the ECOG 1308 trial [17], seek to reduce therapy induced side effects for HPV-positive OPC patients. Different histological methods exist to determine the HPV status. Most frequently used is the detection of HPV induced overexpression of p16INK4a by immunohistochemistry with a reported sensitivity of >90% and a specificity of >80% [18,19]. Non-invasive determination of the HPV status based on routinely acquired radiological imaging may simplify the diagnostic process and reduce costs.

Radiomics refers to the extraction of information from radiological images by applying hand-crafted filters on preselected regions of interest [4]. Segal et al. [5] demonstrated that features in radiological images can be used to reconstruct themajority of the tumor genetic profile. Radiomics data successfully predicted overall survival [6,7], metastases development [8,9] or histological properties [10,11] and may be used as decision support system in clinical practice.

Radiomics approaches to determine the HPV status achieved areas under the receiver operating characteristic curve (AUC) of about 70% to 80%, when tested on external data sets [20,21]. Fujima et al. [22] trained a 2D convolutional neuronal network (CNN) on FDG-PET images to classify HPV status in OPC patients and achieved an AUC of 83% using a transfer learning approach based on natural images from the ImageNet database [23]. But, they did not test their data on an external cohort and excluded images containing severe motion artifacts and tumors with diameter sizes below 1.5 cm.

Although radiomics has proven its potential in medical image analysis, deep learning was shown to be clearly superior in most other computer vision tasks. Deep learning features an end-to-end training without the need to design hand-crafted filters. However, training requires considerably larger patient numbers than typically available in clinical applications. Large 3D-data poses another challenge on the application of deep learning in medical image analysis.

In this work, we investigate deep learning on diagnostic CT-images as a tool to attribute oropharyngeal cancer (OPC) to a human papillomavirus virus (HPV) driven oncogenesis.

Our study was based on 4 different publicly available data sets of the TCIA archive [24]. Transfer learning facilitated deep learning on the relatively small data set size on a CNN derived from the C3D classification network [25], with weights pre-training on the Sports-1M data set [23]. Hussein et al. [26] used C3D to initialize a multi-task learning approach for lung nodule risk stratification. However, their network depended on additional information such as tumor sphericity and texture while we trained our network in a simple end-to-end fashion."

- “TCIA archive” should be written in extenso the first time it appears in the text.
